# Towards distortion-free imaging of the eye

**Phillip Bedggood**⦿*, **Andrew Metha**

Department of Optometry and Vision Sciences, The University of Melbourne, Melbourne, Australia

* pabedg@unimelb.edu.au

## Abstract

The high power of the eye and optical components used to image it result in "static" distortion, remaining constant across acquired retinal images. In addition, raster-based systems sample points or lines of the image over time, suffering from "dynamic" distortion due to the constant motion of the eye. We recently described an algorithm which corrects for the latter problem but is entirely blind to the former. Here, we describe a new procedure termed "DIOS" (Dewarp Image by Oblique Shift) to remove static distortion of arbitrary type. Much like the dynamic correction method, it relies on locating the same tissue in multiple frames acquired as the eye moves through different gaze positions. Here, the resultant maps of pixel displacement are used to form a sparse system of simultaneous linear equations whose solution gives the common warp seen by all frames. We show that the method successfully handles torsional movement of the eye. We also show that the output of the previously described dynamic correction procedure may be used as input for this new procedure, recovering an image of the tissue that is, in principle, a faithful replica free of any type of distortion. The method could be extended beyond ocular imaging, to any kind of imaging system in which the image can move or be made to move across the detector.

## Introduction

Raster-based scanning systems such as scanning light ophthalmoscopy (SLO) and optical coherence tomography (OCT) have become ubiquitous in both clinical practice [1–3] and ophthalmic research [4–6]. Acquired images of ocular tissue are built up over time by raster scanning across the field. Unfortunately, because the eye is in rapid and constant motion [7], no two images look quite the same. This can be described as "dynamic" distortion, which would not occur were the eye stationary. Great care is taken in research applications especially to remove or minimise dynamic distortion where possible [8,9].

We recently described a method to remove, in principle, all dynamic distortion for the scanning light ophthalmoscope [10]. The method was first suggested some time ago [11] and has been confirmed to work with OCT data in three dimensions [12,13]. The method makes a key assumption that the fixational motion of the eye is random in nature, so that if a sufficient number of frames are observed there should be no expected bias in the distortion seen for any particular piece of tissue compared with the rest of the tissue. Where this assumption holds true, this procedure can be expected to remove dynamic distortion entirely. The method has been adopted by several imaging groups [12–17].

warped images in the paper are defined by equations given in the paper. This should make it possible for readers to check and reproduce our work.

**Funding:** Work supported by Australian Research Council (https://www.arc.gov.au/), Discovery Project DP180103393 (PB, AM). The funders had no role in study design, data collection and analysis, decision to publish, or preparation of the manuscript.

**Competing interests:** The authors have declared that no competing interests exist.

Despite its advantages, the dynamic correction method is entirely blind to distortion which manifests in the same way on each acquired frame. We refer to this as static distortion. Such distortion will occur in any ophthalmic imaging system due to marked anisoplanatism which results from the high power of the eye and of the optical systems used to image it, which is one reason that adaptive optics systems are typically limited to 1–2˚ in diameter [18,19]. Major distortion also arises when imaging over an extended field, due to the typically flat shape of detectors compared with the highly curved retina. Static distortions can also occur for raster systems employing resonant scanners due to error in the desinusoiding process [20]; from systematic errors of the hardware [21]; or from eye movement patterns such as nystagmus which are large, repeatable and potentially have periodicity similar to the imaging frame rate such that pseudo-static distortion could result.

To our knowledge the problem of correcting for static distortion does not appear to have been canvased in the ophthalmic imaging literature. It is difficult to state precisely how significant a problem this is, due to a lack of experimental data available on distortion of the eye [22], let alone the combination of eye and imaging system. For the eye alone, schematic eye models can be used to provide some estimate. Over a typical field of view of ~30˚ in a conventional ophthalmoscope pointed at the optic nerve head (15˚ off-axis), distortion from one edge of the field to the other is predicted at ~6% [22]. This is 1.8˚, approximately one third of the diameter of the optic nerve head. Distortion is likely to be even higher a) when including the optics of the imaging system, which lack the excellent wide-field imaging properties afforded by the gradient index of the crystalline lens [19]; b) with an adaptive optics correction in place, as off-axis aberrations typically accrue more rapidly across the field than without adaptive optics [23]; and c) when constructing montages of the retina or imaging with very wide field devices [24], which can span >180˚.

As opposed to the field of ophthalmic imaging, there has been extensive work on the problem of static distortion in the field of "camera calibration" in computer vision research, where it forms part of the broader problem of determining various fixed parameters of an imaging system in a 3D world. Parameters are typically inferred by registration of features within images acquired as the camera pans across a scene, typically by rotating it [25], which has obvious parallels with the constant motion of the eye with respect to a fixed ophthalmic imaging system.

The traditional approach to camera calibration [25] has become ubiquitous in computer vision applications [26]. Although its primary goal is to recover global features such as the camera focal length and viewpoint in the 3D world, it could in principle be applied to ophthalmic imaging. However the following limitations make this approach infeasible:

1. Most commonly this approach employs a known calibration target, but this is not possible to implement in the eye.

2. The method begins by treating the imaging system as a "pinhole" camera that is described by simple geometric transformations. By definition such a camera has no distortion. Distortion is discovered and corrected by using a parametric model for the distorting function, for example a rotationally-symmetric polynomial series. The eye, however, is not rotationally symmetric and, even if it were, off-axis viewing induces a large amount of aberrations which are not [27]; it is also necessary to solve for non-optical sources of distortion as described above, which are difficult to parameterize. A model-based approach is therefore undesirable.

3. The method typically requires that images be acquired from different rotations of the camera, with translations providing redundant information [25]. Whilst the eye certainly *can*

exhibit torsional movement that would rotate the scene in the necessary way as it attempts to maintain steady fixation, the only motion that is *guaranteed* in any image stack is translation of the acquired frames.

4. To our knowledge none of the methods advanced have treated the case of dynamic and static distortions together; only static distortions have been considered.

It should be noted that various developments have been made over the traditional approach cited above, largely solving the first two problems. For example, it is possible to use the natural scene instead of a structured calibration target if the registration can be performed faithfully, i.e. common features can be located across multiple scenes. This is referred to as "self calibration". It is further possible to solve for a complete (pixel-resolved) distortion mapping function rather than to presume some model, which may be termed a "non-parametric" calibration [28]. Various authors have made similar proposals [29–32], which address the first limitation addressed above and sometimes the second, but not the other two limitations.

A form of non-parametric self-calibration has also been proposed in the field of vision science, not for imaging *per se* but for a visual system to compensate for distortion imposed by the optics of the eye and by irregular distribution of photoreceptor cells tiling the retina [33]. In that work it was recognised that movements of the eye shift the retinal scene and that, if the objects in the scene are actually stationary and the system well calibrated, all objects in the scene should move by the same amount. If the calibration is in error, this will no longer hold true. Therefore principled trial-and-error may be used to modify the system's calibration function until the distortion-on-movement is minimised. This is a philosophically similar approach to the one described below, despite the different application. However, major limitations include the difficulty of ensuring convergence of the solution on a global minimum, the potentially large number of image samples required, and a "lowpass" assumption of image formation; these limitations are avoided the present work.

Here, we describe a method that is able to solve for static distortions of arbitrary form by acquisition of images from the translating (and possibly rotating) scene expected for the ever-moving eye. We show further how the previously described dynamic correction procedure can be applied to "clusters" of frames, leaving behind residual static distortion for each cluster. These partially recovered, distorted images can then be corrected by the new procedure such that, in principle, no distortion remains.

## Methods

Broadly, the new procedure requires imaging the same retinal scene across a handful of frames with the eye in different positions (that is, with the images translated and/or rotated with respect to one another). One warped image is selected as a reference, against which the other images are diffeomorphically registered, such that any one of the (warped) images can be transformed to resemble the warped reference. As with the dynamic correction method [10], the approach detailed here takes as input the pixel displacements determined by successful registration; the registration itself is assumed to have been carried out successfully with some existing approach [34]. Here the registration information is used to form a system of simultaneous linear equations, with a variable for each pixel and each mapping between a pair of pixels providing an observation. Solution of this sparse system of linear equations gives the common warp field that was encountered by each frame, as detailed further below. This approach implicitly assumes that the distortion conferred by the eye itself does not change to any appreciable degree over the range of eye positions considered. This assumption is justified given that the approach requires displacements of only a few pixels (each one corresponding

to ~ 0.5 μm in our images), compared with the extent of the isoplanatic zone of the eye which is typically in excess of 300 μm [18].

A note on terminology: distortion maps are defined here as a lookup table of displacements which specify how to build the output image from an input image. For example, if the distortion map in the x direction had a value of -2 at column 100, we would source the data for the pixel at column 100 from column 98 in the input image.

Matlab code and example data are included in S1 File, to assist in implementing the new algorithm.

## Simplified 1D example

Consider the 1D scene shown in Table 1 containing letters of the alphabet "E" through "K", spaced one unit apart. The scene was translated by an amount "t" pixels in the x dimension by either t = 0 units (no shift) at the time of acquisition for image 1, or t = -3 units (a shift to the left) for image 2. The translated scenes are sampled by a 1D detector that is just 3 units long. Anchored to the detector and therefore common to all images is a "static" distortion function f (x), which maps each pixel on the detector to pixel(s) in the translated image scene. In this example the distorting function was chosen to be a parabola, $f(x) = x^2 / 4$.

Suppose that we are tasked with determining the value of the distortion map at the pixel x = +2, using only the two distorted images together with knowledge of the overall translation of the scene. The relevant feature to this exercise, as we shall see, is the letter "H" (bolded in the Table). Other features (for example "FG") are explained further below.

First, consider that the pixel at x = +2 on the detector "sees" a distortion value of $f(2) = 2^2 / 4 = +1$. This means that information at this pixel is sourced from one unit to the right in object space i.e. at x = +3. At the time of capturing the first image, the feature at that location is H (top row of table), and so H is recorded at x = +2 in the acquired image (third row).

Similarly, the pixel at x = 0 on the detector sees a distortion value of $f(0) = 0^2 / 4 = 0$. Therefore data is always sourced from the present pixel (x = 0) in the translated scene. At the time of capturing the second image, the feature at x = 0 is again H, and so H is recorded at x = 0 in the image.

Returning to the task of determining the distortion map, the co-ordinates in undistorted space ($x_U$) of a given feature can be expressed independently in the co-ordinate system of the

**Table 1. Simplified example in 1D.**

| | Position (pixels) | | | | | | | | | |
|---|---|---|---|---|---|---|---|---|---|---|
| | -3 | -2 | -1 | 0 | +1 | +2 | +3 | +4 | +5 | +6 |
| Undistorted space (t = 0) | | | | E | F | G | **H** | I | J | K |
| Static warp f(x) = x²/4 | | | | 0 | +0.25 | +1 | | | | |
| Acquired image #1 | | | | E | FG | **H** | | | | |
| Undistorted space (t = -3) | E | F | G | **H** | I | J | K | | | |
| Static warp f(x) = x²/4 | | | | 0 | +0.25 | 1 | | | | |
| Acquired image #2 | | | | **H** | IJ | K | | | | |
| Registration map (image 2 to 1) | | | | N/A | N/A | -2 | | | | |
| Registered image #2 | | | | N/A | N/A | **H** | | | | |

Example consisting of a 3-unit detector sampling a scene 7 units wide. The scene is imaged twice, after translation of the scene by either t = 0 or t = -3 units. In each case a distortion map f(x) anchored to the detector re-directs the information that would have been intercepted by each pixel. The images resulting from the differing translations together with the common distortion map are shown. The final row provides a look-up table to register image 2 to image 1; it has a single entry, corresponding to the only overlapping feature between the images.

reference image ($x_R$) and in the co-ordinate system of the image to be registered (the moved image, $x_M$):

Reference image: $\boldsymbol{x_U} = \boldsymbol{x_R} + \boldsymbol{f(x_R)} - t_R$

Moved image: $\boldsymbol{x_U} = \boldsymbol{x_M} + \boldsymbol{f(x_M)} - t_M$

In this particular example (i.e. for the feature H), these expressions evaluate as follows:

Reference image: $x_U = 2 + 1 - 0 = +3$

Moved image: $x_U = 0 + 0 - (-3) = +3$

Given the equality of these expressions (i.e. that they both refer to the undistorted co-ordinate space) we can write:

$$f(\boldsymbol{x_R}) - f(\boldsymbol{x_M}) = \boldsymbol{x_M} - \boldsymbol{x_R} - (t_M - t_R)$$
$$= \boldsymbol{d_{M,R}} - t_{M,R}$$

(1)

where $d_{M,R}$ is a vector indicating the output of pixelwise registration between the two images, being equal to the difference in feature location between the two images ($x_M - x_R$). The term $t_{M,R}$ is a scalar representing the full-field translation of the moved image relative to the reference image.

In other words, by mapping a feature in the reference image space ($x_R$) to the same feature in the moved image space ($x_M$), and adjusting for a known amount of whole field translation between the images (t), we obtain the *difference* in the common distorting function [$f(x_R) - f(x_M)$] between two pixels on the detector. We refer to this as a "paired functional difference". The terms on the right-hand side of Eq 1 are measured by registration of the two images (note: whole image translation cannot be reliably inferred from this example with only 3 pixels). The terms on the left-hand side define the (difference in the) distortion map that we are attempting to solve. By obtaining many such paired differences we can constrain the shape of the function (but not its absolute value), as illustrated in Fig 1.

Returning to the example of Table 1, if we acquired other images with varying translations we would possess a set of observations of paired functional differences. For example, with appropriate translations we might discover that:

$$\boldsymbol{f}(2) - f(0) = +1$$

$$f(2) - f(1) = +0.75$$

$$f(1) - f(0) = +0.25$$

This constitutes a system of simultaneous linear equations. In this example, the number of unknowns is equal to the number of observations so that there is no unique solution. However, we are interested only in the *relative* shape of f(x) since adding a constant will merely shift the entire image without distortion. This allows us to set, for example, f(x) = 0 which means here that f(1) = 0.25 and f(2) = 0.75. These match the true values of the function as shown in Table 1.

## Extension to large numbers of pixels

Real images are comprised of many more pixels and hence there will be many paired functional differences to consider. This large set of simultaneous equations can be solved using matrix algebra as shown in Fig 2.

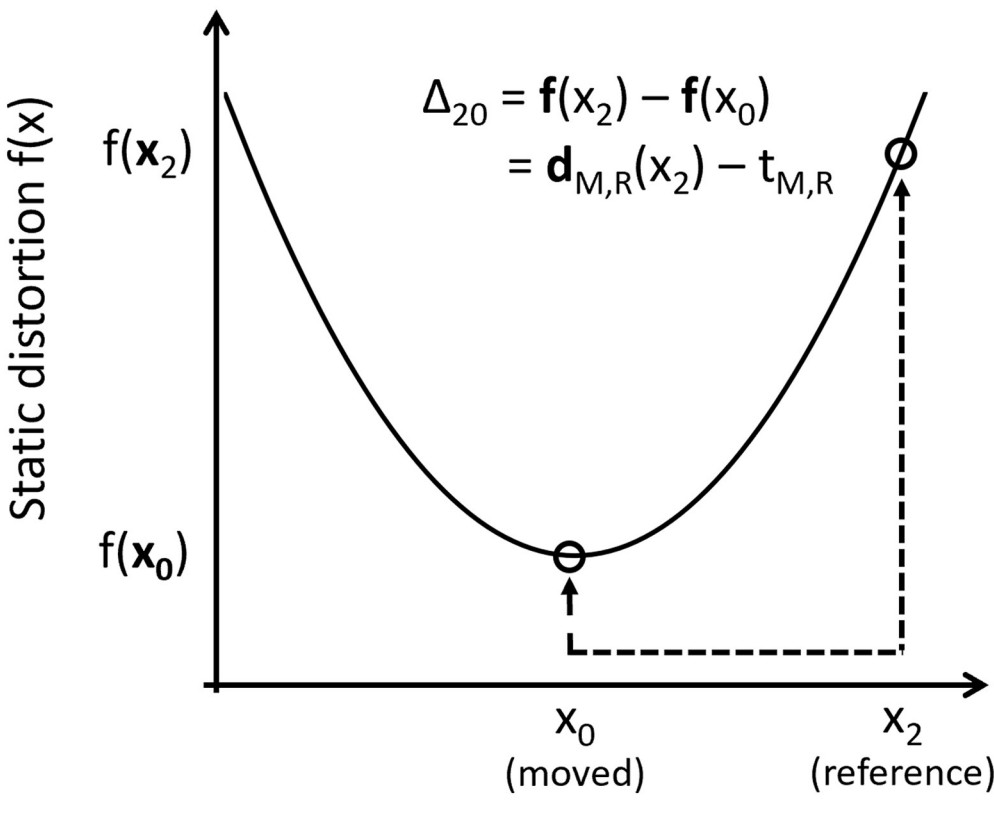

**Fig 1. Illustration of a paired functional difference.** An example static distortion map f(x) is defined over the pixels of the detector (x). Circles represent examples where detector pixels ($x_0$ in the moved image, $x_2$ in the reference image) were determined, by image registration, to have "seen" the same object in different frames as was depicted in Table 1. The value of the distorting function at the two locations is $f(x_0)$, $f(x_2)$. The object in question has shifted between moved and reference frames by an amount ($d_{M,R}$), which is equivalent to the sum of a) the difference in the distorting function $f(x_2) - f(x_0)$, and b) the overall translation of the scene $t_{M,R}$. Therefore, pixel-wise registration between shifted versions of the same scene can be used to build up information about the shape of the distorting function that is common to each frame.

The matrix on the left-hand-side, A, contains one row for each observed functional difference, and one column for each image pixel. Image sets of any dimensionality are handled by appending additional columns, detailed further below. In one dimension there are only two non-zero entries in each row, with coefficients of either +1 or -1 to implement the difference operator (e.g. $f(x_2) - f(x_1)$ is shown in the first row). The vector **f** represents the unknown values of the distorting function to be solved. By multiplying the matrix A with the distortion map **f**, we would obtain the observed image registration data populated in the right-hand-side matrix $\Delta$. Accordingly the unknown distortion map, **f**, can be solved by matrix inversion:

$$\boldsymbol{f} = A^{-1}\Delta \tag{2}$$

The array A is very large. If N is the number of pixels on the detector and M the number of image pairs considered (several images being recommended to ensure the solution is adequately constrained), there will be M*N observations (M*N rows) of the N variables (N columns). For most modern images this will be a very large array whose inverse would be

$$
\underbrace{\begin{bmatrix}
+1 & -1 & 0 & & 0 \\
0 & +1 & 0 & & 0 \\
-1 & 0 & +1 & & 0 \\
& & \cdots & & 0 \\
0 & -1 & 0 & \cdots & +1
\end{bmatrix}}_{\mathbf{A}}
\underbrace{\begin{bmatrix}
\mathbf{f}(x_0) \\
\mathbf{f}(x_1) \\
\mathbf{f}(x_2) \\
\cdots \\
\mathbf{f}(x_n)
\end{bmatrix}}_{\mathbf{f}}
=
\underbrace{\begin{bmatrix}
\mathbf{d}_{M,R}(x_0) - t_{M,R} \\
\mathbf{d}_{M,R}(x_1) - t_{M,R} \\
\mathbf{d}_{M,R}(x_2) - t_{M,R} \\
\cdots \\
\mathbf{d}_{M,R}(x_n) - t_{M,R}
\end{bmatrix}}_{\mathbf{\Delta}}
$$

with column headers $x_0 \quad x_1 \quad x_2 \quad \cdots \quad x_n$ above matrix $\mathbf{A}$.

**Fig 2. Linear algebra representation required to solve for the static distortion function f(x).** A sparse array (A) populated by entries of ± 1 is used to perform pairwise difference operations between image pixels $x_i$ for the distorting function f(x). The resulting "paired functional differences" are given in Δ. This array is equivalent to the vector required to register the two images ($d_{M,R}$), after accounting for the scalar whole-field translation between the images t. The unknown values f(x) can then be solved by pseudoinversion of the large, sparse matrix A. The highlighted row corresponds to the example depicted in Table 1 and in Fig 1.

prohibitively resource intensive to compute. However, it is not necessary to compute the inverse directly. Since the matrix is sparse there exists an efficient solution for the expression $A^{-1} \Delta$ through minimization of least squares. We implemented this using the Matlab function "lsqminnorm" (MATLAB R2017b, The Mathworks, USA). Although this function has additional benefits in selecting an appropriate solution to under-constrained problems, we found that it was necessary to constrain the problem by other means as detailed below.

## Constraining the solution

With only two images as illustrated for simplicity above, the solution is poorly constrained. In a well-constrained solution, if two pixels are not directly related it would still be possible to learn the functional difference between them by tracing a series of paired functional differences across the image. Multiple image pairs are generally required to ensure that all pixels are relatable in this way to all other pixels, constraining the solution. When considering translations alone, typically several translations are required which should vary in degree and in direction in order to provide sufficient information. If these conditions are not met, translations alone can result in non-converging "streams" of pixel relationships across the image, such that some parts of the distorting function are not relatable to other parts.

Interestingly, when rotation is included, this problem of non-relating "streams" dissolves because pixel relationships no longer travel in straight lines across the image. Extension of the method to handle any type of affine transformation, as opposed to just the translations illustrated above, is covered below.

## Handling non-integer displacements

The simplified example above for feature "H" above (Table 1) considered direct 1:1 correspondences between the information seen by each pixel. In reality, precise registration will often

yield fractional displacements, such that mapping is not 1:1 (e.g. in Table 1, "FG" was recorded at x = 1 in image #1).

In principle this problem could be solved to arbitrary precision by upscaling the images, however this is impractical because the solution is already very resource intensive. Instead, we implemented a form of 2x2 anti-aliasing which produced satisfactory handling of fractional displacements. Under this scheme, the negative entries in the coefficient matrix, which were -1 in the simplified example, are distributed between the 4 pixels bounding the fractional pixel address, weighted by their proximity to that address. The sum of the coefficients for these 4 pixels remains -1, and the "reference" pixel is left unchanged at +1. The appropriate pixel weightings for each of the 4 bounding pixels were calculated by the product of independent weightings for x and y proximity, as follows:

$$W(x_a, y_b) = (1 - |x - x_a|) * (1 - |y - y_b|)$$

The weightings calculated this way sum to unity for the 4 pixels, and dictate the proportion of the given object's intensity that will be recorded in each of the 4 locations.

## Handling distortion in more than one spatial dimension

The examples above refer only to displacements in the x direction. Real 2D or 3D image data can/will be distorted in the other dimensions as well. For each pixel (or voxel) in the image, a separate variable is needed for each dimension. For example in a 2D image with N pixels, one image pair will require 2N columns and 2N rows in the coefficient matrix A. Each additional image pair considered requires an additional 2N rows.

In the case of pure translation, the array just described is unnecessarily large because the x and y (and z) problems could each be treated independently. The reason for structuring the array in this fashion is that in the case of other affine transformations of the scene, such as rotation, there is "cross talk" between the cardinal axes which must be accounted for, as described below.

## Affine transformation of co-ordinates

The rationale given above concerns only translations between recorded image scenes. However, other affine transformations of the scene could be observed, with image rotation due to torsional movement of the eye being a known common problem in registration of ophthalmic imagery [35,36]. Affine transformations can be handled as follows:

$$f(\boldsymbol{x_R}) - f(\boldsymbol{x_M}) = \boldsymbol{x_M} - T_{1,1} * \boldsymbol{x_R} - T_{1,2} * \boldsymbol{y_R} - T_{1,3}$$
$$f(\boldsymbol{y_R}) - f(\boldsymbol{y_M}) = \boldsymbol{y_M} - T_{2,1} * \boldsymbol{x_R} - T_{2,2} * \boldsymbol{y_R} - T_{2,3}$$

(3)

Here the term T indicates a standard affine transformation matrix which can implement any combination of translation, rotation, scale and shear. The numbers in subscript indicate the row and column entries in the transformation matrix. For the example of rigid body transformations (translation and rotation alone), the transformation matrix (T) would appear as:

$$T = \begin{bmatrix} cos\theta & -sin\theta & tx \\ sin\theta & cos\theta & ty \\ 0 & 0 & 1 \end{bmatrix}$$

(4)

Hence both the equations in x and in y have terms which account for the effect of rotations and translations. For example in the x-direction, it can be seen that in addition to accounting

for a whole-field image shift $T_{1,3}$ or *tx*, we must also take account of the rotation by transforming the found co-ordinates in the reference image ($x_R$, $y_R$). Importantly, the paired functional difference in the x-direction must now consider the y-location found in the reference image–there is interaction between the dimensions which must be accounted for as shown.

It will be shown below that using the above transformations, rotation of the scene does not present any difficulty for the proposed algorithm. However, rotations must be estimated accurately for input to the algorithm; this can be a challenging task depending on the particular imaging modality [35,36]. Image registration is detailed in the next section.

## Image registration

As with our previous work, the proposed algorithm should be conceptualised as a "meta-registration" procedure, meaning that it takes as input the results of successful image registration rather than carrying out the registration itself. Hence to demonstrate proof of principle it is not necessary to carry out "real" image registration; idealised values that were used to create the imagery were used. This saves computation time, but more importantly it avoids failure resulting from the image registration itself as opposed to a flaw in the proposed algorithm. Such failures would be hard to detect without manual review of the thousands of images simulated here.

Nonetheless, it is important to demonstrate that the approach can be integrated with an image registration pipeline operating blindly on real data. Accordingly, we have implemented "real" registration in the example provided in S1 File. This involves initial estimation of the full-field shifts required to broadly register images in the stack, which was achieved with a standard Fourier domain cross-correlation approach. For estimation of distortion maps we employed the "demons" diffeomorphic registration algorithm [34], available in MATLAB through the Image Processing Toolbox (MATLAB R2017b, The Mathworks, USA). This algorithm was required rather than the "strip" registration conventionally employed for raster-scanned data, because static distortions will rarely be entirely directional so as to warrant a strip-based approach. The robustness of the proposed algorithm to errors in image registration may be explored by increasing the noise parameter included in S1 File.

## "Ground truth" data

As in our previous work [10], "ground truth" images were obtained from by imaging the cone photoreceptor mosaic in a young, healthy subject at high speed (200 fps; frame exposure 2.5 ms) with a flood-illuminated adaptive optics ophthalmoscope and an imaging wavelength of 750 ± 25 nm. Images were acquired over a 7.5 mm pupil after adaptive optics had reduced the root-mean-square (RMS) wavefront error to 0.05 μm or less, i.e. below the diffraction limit. The ground truth data were montaged from multiple imaging locations temporal and within 2° of the foveal centre. This allowed a relatively wide field to be imaged without noticeable degradation in quality towards the edge of the field. Images of the photoreceptor mosaic in general provide an excellent example to evaluate the presence of image distortions due to the presence of high frequency information distributed uniformly across the image. In this specific dataset, there is negligible static or dynamic distortion because:

- Images were acquired with a flood-illuminated system such that all pixels in a frame are acquired simultaneously. Dynamic distortions encountered across the field by raster sampling the moving eye are therefore avoided.

- Image data presented are from a 512x512 pixel area which corresponds to approximately 0.9° in diameter. This is well within the lower bounds of the isoplanatic patch based on

previously published estimates [18], hence, there should be minimal optical distortion encountered due to off-axis aberrations of the combined imaging system of the eye and ophthalmoscope.

As in our previous work, the ground truth data were used here to simulate imagery acquired in the presence of distortion. The fidelity of distortion recovery was assessed by calculating the Pearson correlation in image intensity between the recovered and ground truth images. Applied distortion was either static alone, which could be produced by a "flood" illuminated system or by a scanning system operating at high frame rate or on a stationary eye; or a combination of static and dynamic distortion, which could be encountered with a scanning instrument in the presence of fixed sources of distortion. It should be noted that our simulation does not consider common differences which may be found between flood or scanning image modalities, for example regarding shot noise or speckle characteristics which are not expected to affect distortion *per se*.

It should be noted that the algorithm described here operates on image registration data, rather than on the images themselves. As described above (see section "Image Registration"), for the majority of data presented here we carried out idealised registration does not consider the content of the images themselves; the images are merely useful tools to illustrate the degree of distortion. Hence, the use of cone photoreceptor images does not diminish the applicability of the proposed method to other types of images. The uniform, high spatial frequency content of such images does provide an excellent canvas in which to make comparisons, compared for example to images collected by other modalities which may lack "interesting" information across broad regions of the image, potentially masking residual distortion.

This project was carried out in accordance with the principles expressed in the Declaration of Helsinki, and approved by the University of Melbourne Human Ethics Committee. Written informed consent was obtained from the subject prior to testing.

## Simulating full-field movements

Simulating overall motion of the eye was accomplished by applying translations and rotations to the ground truth retinal scene, generally differing in degree and direction to provide complementary information as described above. Specific movements applied are detailed in the Results. This approach builds upon our previous work where only translations of the image were considered, and where data were limited to those eye movements which happened to occur during a particular imaging sequence.

Since one cannot select what movements a real eye undergoes, we tested the new method with a large number of simulated runs, with the parameters for rigid body movement randomly drawn on each run. For this exercise the translation parameters were chosen from the range ± 10 pixels and the rotation parameter ± 5˚. As described above under "Constraining the Solution", it is necessary to ensure that images selected contain "unique" displacement information. We therefore applied somewhat arbitrary partial constraints to the translation of the retinal scene: translations each differed in magnitude by at least 2 pixels, and in direction from the origin by 45˚ or more. These constraints are by no means optimal, but could serve as an initial guide for selecting images from real data sets.

## Simulating distortion

As described in the Introduction, it is difficult to know how much and what sort of static distortion is present in ophthalmic imaging systems. Our general philosophy was therefore to

simulate distortion of greater magnitude and with more complex structure across the field than we believe might be encountered in practice. This should serve to convince the reader of the robustness of the proposed method.

To test static distorting functions of relatively arbitrary shapes, we applied 2D sine functions (in x and y) of varying amplitude, phase and frequency as follows, where x and y ranged from 0 to 2π over the full image array:

$$D_x(x, y) = A_x \sin(f_x x - \varphi_x)$$
$$D_y(x, y) = A_y \sin(f_y y - \varphi_y)$$

(5)

There were accordingly 6 parameters set for each distorting function. It should be noted that the horizontal distortion in Eq 5 varies only in x, and the vertical distortion only in y. This makes it easier to visualize the results because the distortion map can be meaningfully plotted in one dimension. However the method is not limited to application in this way and works just as well for arbitrary distortions. Similarly, the method works just as well for simpler distortions, for example the rotationally symmetric distortions expected for a centred, high power system. We chose to present results from sinusoidal distortions here, in part because they look sufficiently bizarre to convince the reader of the broad applicability of the method, and in part because of the well-known Fourier decomposition theorem which allows arbitrary shapes to be reconstructed from a sinusoidal series.

The above parameters allowed us to test distortions of high amplitude, and with unusual structure (e.g. asymmetric, minimum not coinciding with the array centre, multiple cycles across the image) to ensure robustness of our approach. In the Results presented below, the amplitude parameters lay in the range ± 5 pixels, the frequency parameter in the range 0 to 5 cycles, and the phase parameter in the range 0 to 2π.

We also present data in which dynamic and static distortions are combined. Dynamic distortions were implemented by superimposing full-field movement of the scene with 2D sinusoidal distortion following Eq 5, with new free parameters randomly drawn for each frame. To mimic the behaviour of raster systems whereby minimal dynamic distortion is expected along the "fast" scan axis, for dynamic distortion we modelled variation of the distorting functions only in the y direction (i.e. $D_x(y)$ and $D_y(y)$). This assumption could be violated during very fast (saccadic) eye movements, however, we do not consider such movements here due to the irrecoverable motion blur which would result.

Although the sinusoidal distortions modelled do not have particular "real life" analogues (the eye is unlikely to follow sinusoidal motion), this approach allowed simulation of distorting functions that varied widely in degree and spatial structure in an unbiased manner. There is also no correlation in distortion applied between one frame and the next as would occur for a real eye. This helps to avoid a potential criticism of our previous work [10], where it might be supposed that the eye movement data underlying our simulations happened to be a fortuitous set for the purposes of image recovery. We note further that our previous work [10] and its subsequent application by others [12–17] have already confirmed the dynamic correction method to correct for dynamic distortion in the presence of realistic eye movement patterns. In the modelling below, we will show that as long as a satisfactory dynamic correction is obtained, the new static correction procedure can then be applied to recover images that are free of both kinds of distortion. The significant distortion entailed by the present modelling can be appreciated by the sequence of 100 consecutive frames shown in S1 Fig.

## Dynamic correction method

Although described in full elsewhere, it is worthwhile to give a brief overview of our previously published method for correcting dynamic distortions [10]. These are generated dynamically in raster-based imagery because eye movements occur during the sequential acquisition of image points. The method relies on the assertion that, given the stochastic nature of fixational eye movements, the expected value of distortion "seen" by one part of the retina should not differ from that expected for any other part of the retina. In other words, the bias in apparent displacement for each piece of tissue should trend towards zero when many frames are considered. Accordingly, the local distortions accrued within any particular frame can be discovered by registering each piece of tissue therein across a large number of frames; if it is found that the average displacement for that tissue differs from zero, this must result from the distortion inherent in the frame in question, such that the net bias does sum to zero.

The assumption of zero bias described above must be violated in the case of systematic error which produces static distortion, present on every frame. In that case, assuming that the bias is zero should be expected to result in the full degree of underlying systematic bias remaining within the "recovered" frame. This underpins the strategy advanced in the current work, whereby dynamic recovery is first applied in order to produce a handful of frames which all suffer from a common systematic bias.

## Array size

Similar to our previous publication, initial dimensions of simulated images were 400 x 400. However, the high memory requirements of the proposed algorithm left us unable to solve arrays of this size on a laptop computer with 16 GB RAM. We therefore clipped to the central 200 x 200 when supplying images and distortion maps to the DIOS algorithm. This limitation and potential solutions to it are expanded on further in the Discussion.

## Results

### Static distortion only

Fig 3 shows example scenes of the cone photoreceptor mosaic with the eye in slightly different positions and with 2D static distortion applied (i.e. the warping function is identical in each frame). The true distorting functions in x and y are plotted in Fig 3). Four frames were simulated with mean position given by the coloured crosses; two example frames are shown (Fig 3A and 3B). There was no rotation between images in this example. Following registration of the images to the first frame using the "demons" algorithm as described above, the registration information was passed to the DIOS algorithm. Example images recovered are shown in Fig 3D and 3E, giving strong correspondence to the ground truth (Fig 3C). Similarly the recovered distortion function is very close to the function that was used to generate the images (Fig 3F).

Fig 4 illustrates two potential pitfalls of the proposed algorithm. The same warped examples as Fig 3 are shown, however now only two of the frames were input, which are offset from one another in the vertical direction only (Fig 4C, crosses). It can be seen that strong distortion remains in the x direction (Fig 4D and 4E) and that a flat distortion profile was estimated in this direction (Fig 4F). This demonstrates that images need to be offset from one another obliquely if they are to facilitate recovery of image distortion in 2D (and displacements would presumably be required in the third dimension as well in the case of 3D data). For this reason we have termed the approach "DIOS", or Dewarp Image by Oblique Shift. The other pitfall illustrated here is seen in Fig 4F, where even in the y-direction "wiggles" are apparent in the shape of the recovered distorting function. This illustrates that the solution is poorly

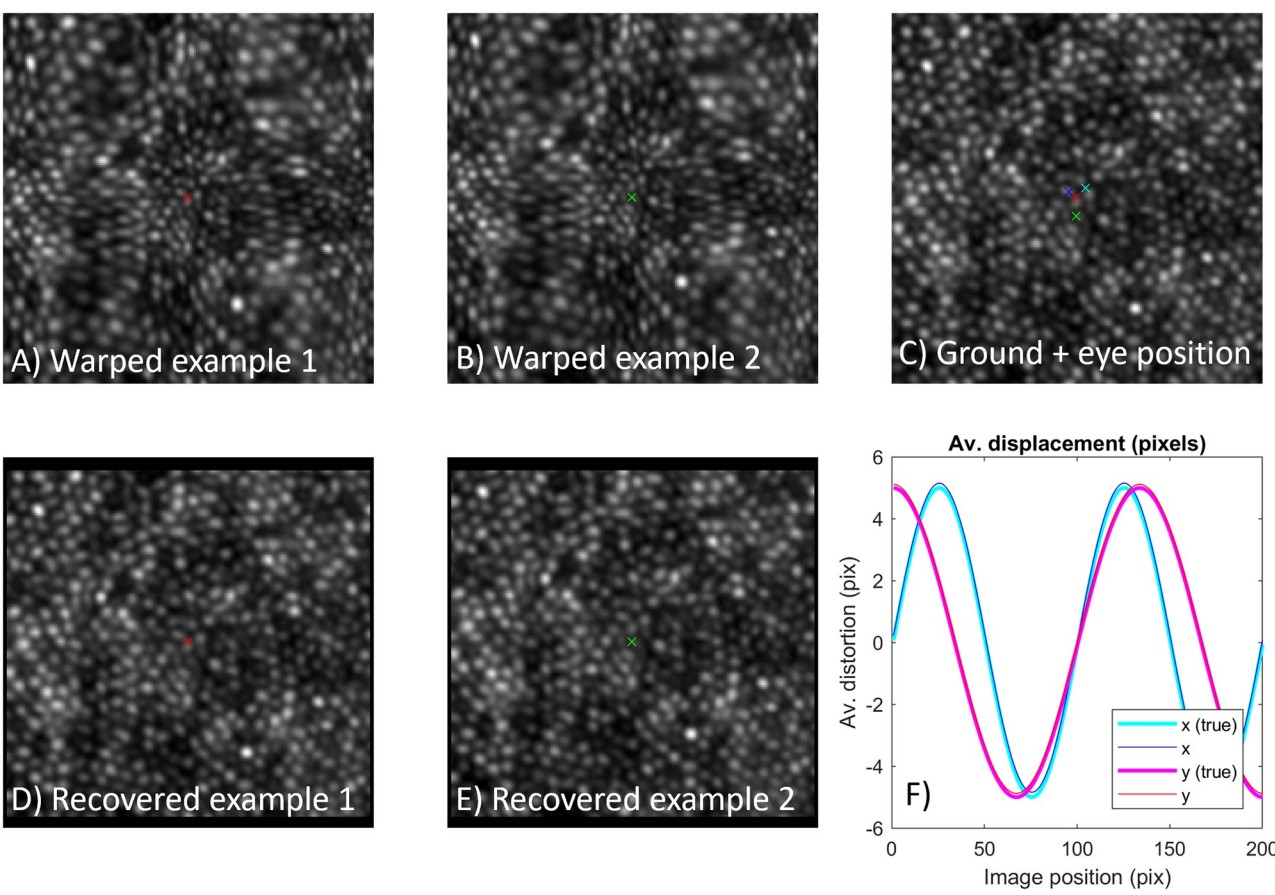

**Fig 3. Example correction of static distortion in 2D.** Four distorted frames were generated with the eye at different positions corresponding to the coloured crosses. Example imagery is shown (A, B) which appears highly warped compared to the ground truth data (C). Nonetheless, images are recovered with high accuracy (D, E). The profile of recovered distorting functions is plotted (F), showing strong similarity to the true distorting profiles.

constrained, which occurs because only a single frame pair was input to the procedure. As described above this leads to "streams" of pixel relationships across the image, such that certain pixels cannot be related to other pixels in the image.

Fig 5 demonstrates that the proposed procedure can properly handle rotations of the scene, that is, torsional movements of the eye. Fig 5 illustrates data simulated in the same way as Fig 3, but with rotations spanning ± 30˚. It can be seen that, as for Fig 1, recovered images and distorting functions appear very similar to the ground truth, indicating that rotation was adequately addressed.

To demonstrate that the method can handle distortions of arbitrary form, we modelled 1,000 static distorting functions whose parameters (Eq 5) were drawn uniformly from the full range described above (see section "Simulating distortion"). Given the computation requirements for larger arrays, here we recovered only the central 100x100 pixel region. The simulation parameters supported extreme distortions (the most distorted example is shown in Fig 6A). For each simulated run shown in Fig 6, we "acquired" 4 frames with the eye having been translated and rotated by random amounts (see section "Simulating full-field movements").

The outcomes of these 1,000 simulated runs are plotted in Fig 6. Even for the worst case simulated (Fig 6A), the recovered image was very similar to the ground truth (Fig 6B; correlation > 0.99). Fig 6C shows the initial (red) and recovered (blue) correlation for all 1,000

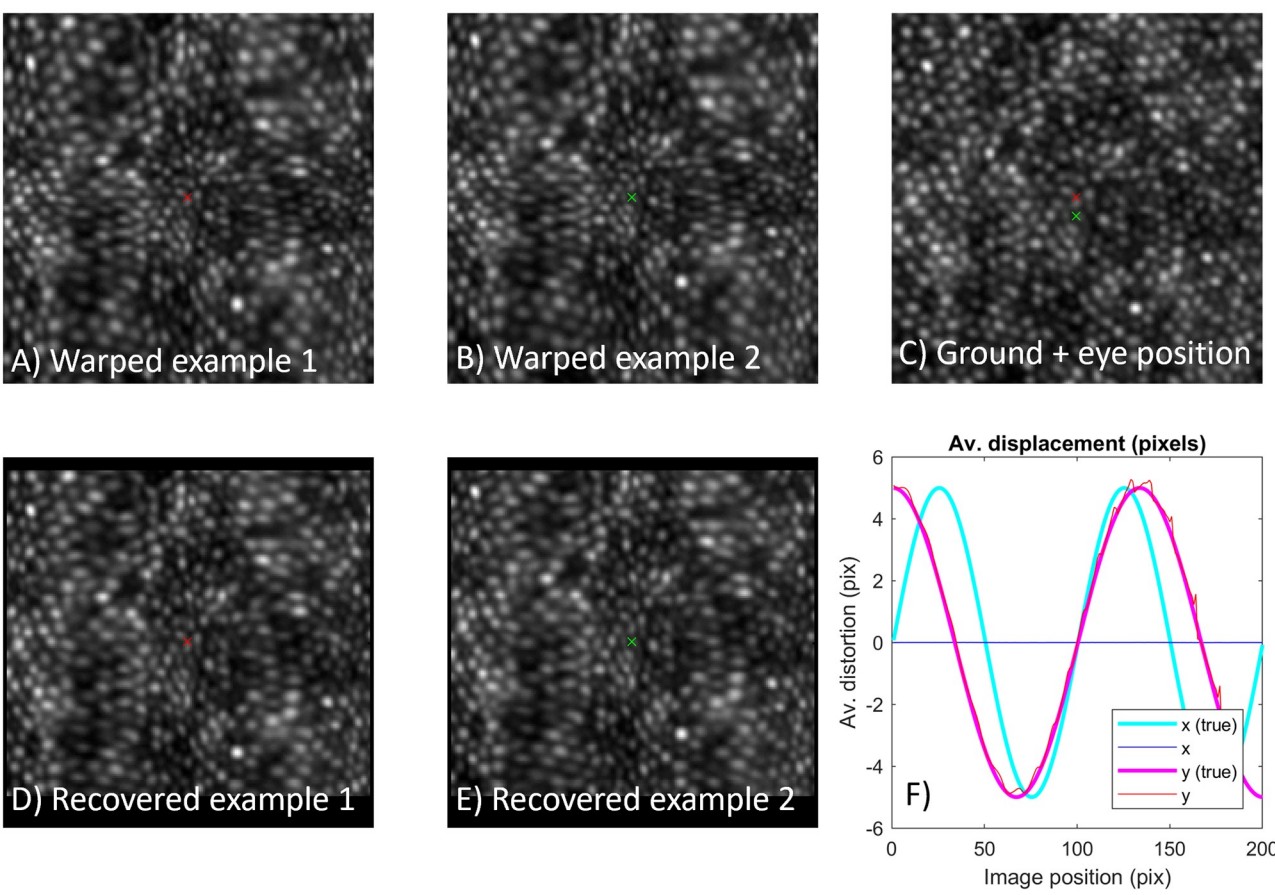

**Fig 4. Example failure to correct for static distortion in 2D.** Two distorted frames were generated with the eye at different positions corresponding to the coloured crosses. Generated images (A, B) and ground truth (C) correspond to those in Fig 1. Recovered images are still seen to be highly warped in the horizontal direction (D, E). This occurs because there is no information provided regarding differences in the horizontal direction (flatline in F). In the vertical direction, the use of just two frames fails to constrain the solution, producing a "shakey" appearance to the recovered distortion profile (red line in F).

simulated runs, confirming excellent similarity to the ground truth image over the vast majority of simulated runs (only 17/1000 runs showed correlation below 0.99).

## Static and dynamic distortion

The DIOS procedure may be applied in sequence with the dynamic recovery procedure previously described [10] to recover images that are, in principle at least, distortion-free. To illustrate this process we modelled combined dynamic and static distortion, with static distortion as shown in Fig 7 (black). This is a destructive example because all static distortion coincides in direction with that of the dynamic distortion (the direction of the "slow" scan axis). Each frame was also warped dynamically, with distortion in both x and y directions generated from Eq 5 with frequency and phase parameters drawn from the full range described above (see Methods section "Simulating distortion"). The amplitude of the distorting function here was fixed at 5 pixels to ensure dynamic warp of all frames (i.e. contrary to real data, there is no single frame which could match the ground truth). Example frames warped by both static and dynamic distortion are shown in Fig 8A and 8B.

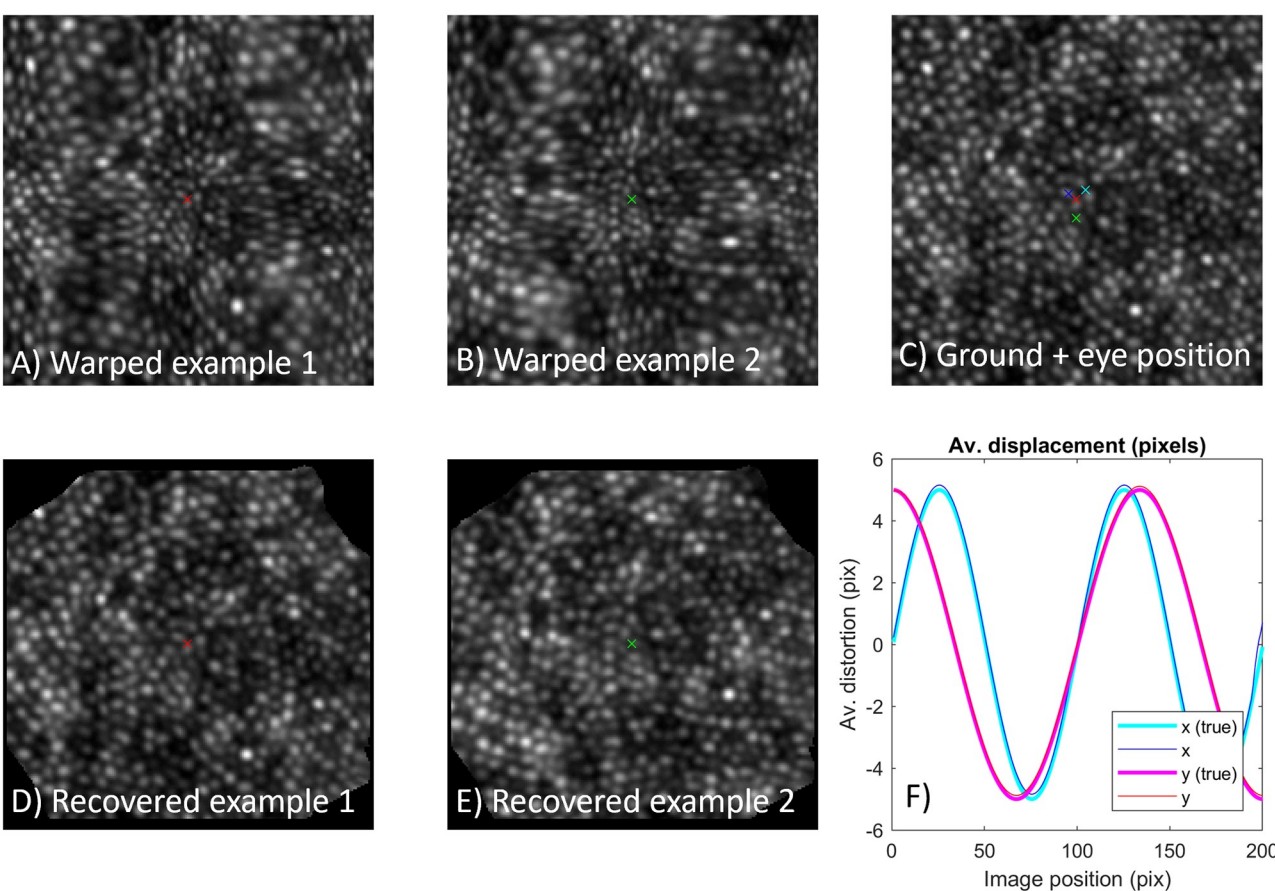

**Fig 5. Example correction in the presence of rotation.** Four distorted frames were generated with the eye at positions as for Fig 1, however now with rotation spanning 50˚ between the 4 frames. Example images are shown (A, B) which are rotated by 30˚ with respect to each other. Image A is the same as that for Fig 3. Both example images are recovered with high accuracy (D, E) and recovered distorting profiles (F) match the ground truth, demonstrating robustness to image rotation.

Distorted frames were generated in 6 "clusters", with each cluster having slightly different mean gaze position (up to 10 pixels) as required by the DIOS algorithm. Each cluster also had a different torsional component (up to 10˚), to demonstrate robustness to image rotations. Clusters consisted of 1000 frames to ensure that bias resulting from dynamic warp would trend close to zero. Unlike the simulation of random static distortions that was shown in Fig 6, there was no special requirement imposed on gaze position within a cluster, other than that it be randomly drawn from a normal distribution (scaled so that max. amplitude within a cluster was 50 pixels). The simulated eye movement pattern within each cluster was independent from the other clusters. The eye position for all frames and all clusters is indicated in Fig 8C (coloured dots; a different colour for each cluster), overlaid on the ground truth image.

The dynamic recovery procedure was run on each cluster independently, producing a single recovered frame (examples shown in Fig 8D and 8E). These frames appear qualitatively to be much less distorted than the example input frames (Fig 8A and 8B), however significant distortion remains (e.g. Fig 8D has a correlation of 0.67 to the ground truth image). This can be appreciated visually by comparing an example constellation of 5 cells (red polygons) with the same constellation from the ground truth image (yellow polygons). Applying dynamic recovery to all 6 clusters provided 6 frames for input to the DIOS procedure, which returned the

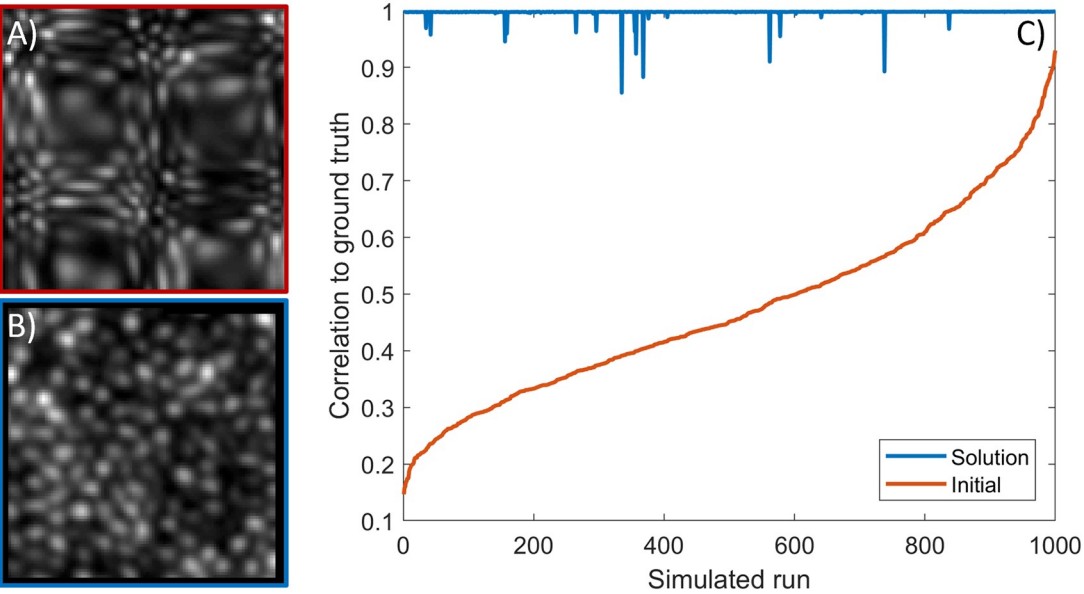

**Fig 6. Outcomes from 1000 different static distortion maps.** On each run a random 2D distorting function was generated, and sampled at 4 randomly selected gaze positions (see text for details). A) Shows the worst simulated image, i.e. the one least correlated with the ground truth. B) Shows the recovered image for A. C) Plots the baseline (red) and recovered (blue) correlation with the ground truth for all 1000 runs. Only 17/1000 recovered images failed to reach a correlation above 0.99.

solution plotted in Fig 7 (cyan). The distortion required to faithfully render the ground truth image without error is shown in blue; their strong overlap indicates that the solution was accurate (RMS error = 0.18 pixels). This can be further verified by the similarity with the ground truth image of Fig 8C (correlation = 0.991; recovering the images drawn from other clusters gave near-identical results with the worst correlation at 0.986) and by the reference constellation (overlap of yellow and red polygons in Fig 8F).

Interestingly, in this example the required correction differed substantially from the underlying static distortion (compare cyan and magenta curves in Fig 7A). This means that the bias remaining after dynamic recovery was not simply equivalent to the underlying static distortion, indicating some interaction effect. To investigate this, we zeroed the vertical component of within-cluster eye movements and repeated the simulation. The solution obtained is shown by the red curve in Fig 7, which now closely resembles the underlying static distortion in magenta (RMS error = 0.19 pixels). This result suggests that panning the tissue across different regions of the static distorting function conferred a pseudo-dynamic distortion. Despite this, the combination of the dynamic and DIOS procedures seemed able to recover the solution as was shown in Fig 8. To learn whether the dynamic, sinusoidal distortion also interacted with the static component, we repeated the simulation with zero dynamic distortion (and with the original eye movements). This resulted in the black curve of Fig 7, which is highly similar to the underlying function that was applied, indicating minimal interaction (RMS = 0.17 pixels).

## Discussion

We have demonstrated a new method to estimate and correct for static distortions in an imaging system. The method is particularly suited for ophthalmic imaging because: a) no calibration target is required, which is not available in the eye; b) no model is assumed for the distorting function; c) the method requires only full-field movement of the scene which is guaranteed by the incessant motion of the eye; d) the method can handle torsional movements

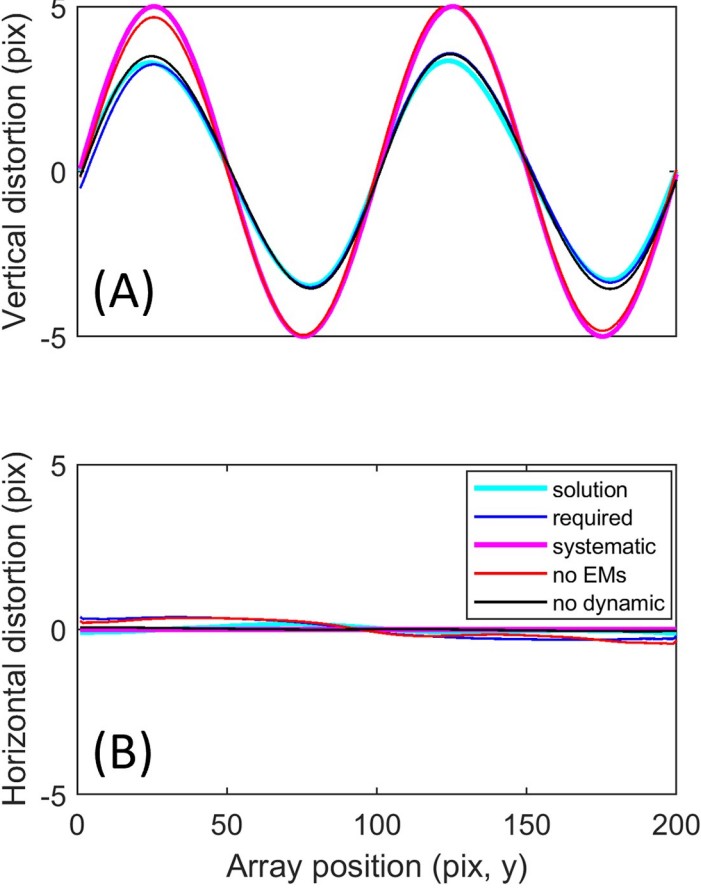

**Fig 7. Static distortion profiles applied and recovered in the presence of dynamic distortion.** Six clusters of 1000 frames were generated (see text and Fig 8). Average distortion along the y array direction is plotted for vertical distortions (A) and horizontal distortions (B). The recovered solution is shown in cyan, matching the required solution in blue. The true underlying static distortion is shown in magenta, but this is only the correct solution when the eye did not move vertically (red). Black shows the solution when there were only eye movements (no sinusoidal warp applied).

which can occur for the fixating eye, but does not rely upon them; e) the method can be used in series with a previously published approach to correct dynamic distortion as well, which is encountered by modern ophthalmic imaging modalities.

The method takes as input the pixelwise registration maps between a series of images and a reference image, and assumes that the affine displacement (e.g. translation, rotation) between images can be accurately inferred despite the presence of image warp. This affine information is used to "correct" the registration maps, such that it provides information on the difference in the distorting function between pairs of pixels on the detector. If multiple images are acquired with the eye having moved in different directions by different amounts, this series of observations can be used to solve for the relative shape of the distorting function. The method is blind to the absolute value of the distorting function, that is, to a fixed prismatic shift applied to all images. Such shifts merely move the image rather than producing distortion. The method would also be blind to uniform scaling (magnification) of the entire image set, unless the true amount of eye movement is known.

Our modelling suggests that the method is capable of arbitrary precision, with residual errors a small fraction of a pixel. However, there are some caveats to this. A sufficiently large

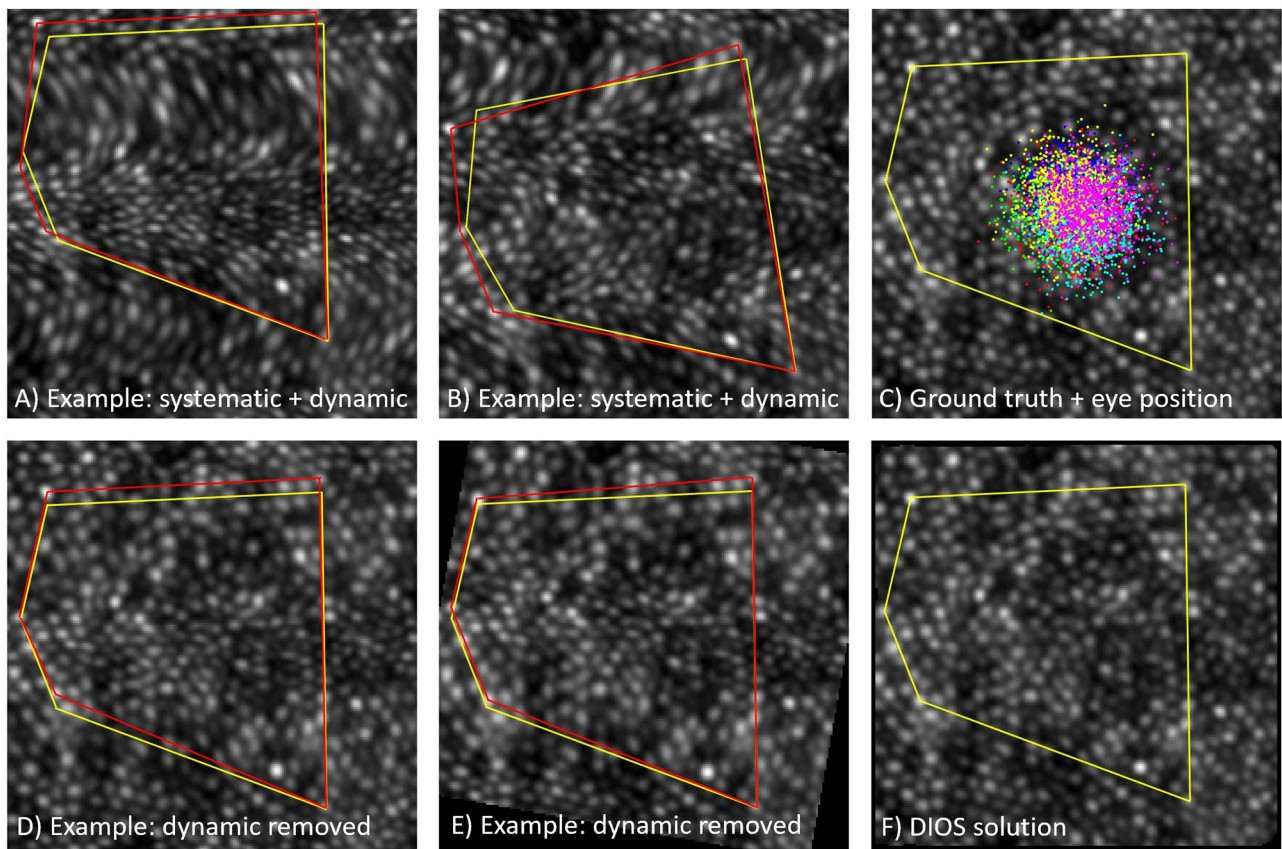

**Fig 8. Correction of simultaneous static and dynamic distortion.** A) Example image from a cluster of 1,000 frames, warped by both static and dynamic distortion. B) Example image from a second cluster; only dynamic distortion differs from A. C) Ground truth image, overlaid with the gaze position for six 1,000 frame clusters (each cluster a different colour). D) Output of the dynamic recovery procedure on the cluster corresponding to A. E) Output of the dynamic recovery procedure on the cluster corresponding to B. F) Output of the DIOS solution, running on the six outputs of dynamic recovery (one for each cluster), examples of which were shown in D and E. Red polygons: a constellation of cells to aid visualization of correction for distortion. Yellow triangles: the same constellation of cells in the ground truth image, shifted and rotated as needed to minimise error.

number of frames (more than two) must be acquired, with the eye having moved in different directions by different amounts, in order to adequately constrain the solution. Where this is not the case, there are effectively "islands" of connected pixels produced within the image. A simple example where this occurs would be when the translation between images is vertical only, as in Fig 4; the "match" for each pixel will then always be found above or below it, but never to the right or left. This means that each column of the image effectively becomes an independent system of equations to solve. Because an arbitrary offset must be set for each system of equations (as the algorithm only solves for relative differences), these islands will invariably end up at different absolute displacement values, producing a "bumpy" appearance to the solution. One limitation at present is that we can offer no guidance as to whether a particular set of affine transformations will provide sufficiently unique information to constrain the solution; the output needs to be reviewed for consistency.

Any correction applied to real images must use real image registration, which will be subject to some degree of error. Although the least squares nature of the solution is likely to be resilient to such errors, we have explored this only briefly (see S1 File), wishing here primarily to demonstrate in principle that the method itself does not impose any significant limitation to accuracy. Note that consideration of error due to inaccurate image registration encompasses errors

which may arise due to frame-to-frame variations in intensity. Such variations could be caused by factors such as light-evoked fluctuations in reflectance, the oxygenation state of blood, shot noise, non-uniformity of illumination, vignetting, etc. Where these do not cause error in image registration, such intensity variations will not affect the quality of the recovered distortion maps, because the algorithm presented operates only with registration information rather than intensity information.

To correct for both simultaneous static and dynamic distortion, one strategy is to first correct for dynamic distortion within a given "cluster" of frames, and to pass the output from several such clusters to the DIOS procedure. The mean position of the eye for each cluster needs to be somewhat different in order to provide the required offsets for the DIOS algorithm; this could be achieved in a real dataset by judicious separation of a full set of frames, acquired at the same nominal position of fixation, into pseudo-clusters that each carry the desired mean position. There is no requirement that the clusters need be composed of entirely different frames or that they be contiguous in time, and differences in mean position required need only be a handful of pixels ($<= 10$ in our simulations). In our modelling we used a very large number of frames in each cluster to meet the fundamental assumption of the dynamic recovery procedure (i.e. that bias in displacement for each particular piece of anatomy is close to zero). Where this assumption is not met (e.g. if too few frames are considered), artifactual distortion could remain that will not be common with other clusters of frames, which could produce error in the DIOS solution.

It may at first appear surprising that the output of the previously described dynamic correction procedure would combine well with the newly proposed static correction procedure. However, this marriage is assured by the large number of frames that we included within each cluster. This means that the interaction occurring between systematic and dynamic warp within one cluster should be expected to be nearly identical to any such interaction taking place in another cluster. In other words, the residual static error should not differ appreciably between clusters, given the large number of frames within each cluster. This satisfies the conditions for successful application of the DIOS procedure (a distortion function that is common to all clusters), such that near-perfect correction should be expected. It was noteworthy, though, that the residual static correction obtained was not simply equal to the ground truth static correction applied. Further work should explore the interaction between systematic and dynamic sources of distortion, which is likely to depend on the degree and spatial frequency of each type of distortion.

One alternative to the use of clusters with very many frames may be to use a handful of single frames where the eye has undergone ostensibly the same pattern of movement within each frame. Where this occurs, dynamic distortion will be common between each image. Together with the common static distortion, the image could be considered to suffer from common *pseudo*-static distortion that can then be corrected by the DIOS procedure alone (i.e. without first requiring the dynamic recovery procedure). We have confirmed this to work in simulations (data not shown), however the question remains as to how many frames one would need to acquire in order to possess a handful with sufficiently similar eye movements to support this approach.

A limitation of the approaches advanced above to correct for static and dynamic distortion is that one may not be able to perform a "once-off" system calibration, or to solve for eye movements with complete accuracy, due to interactions between static distortion and dynamic sources of distortion (e.g. the difference between red/magenta and the other curves in Fig 7, showing respectively that the underlying static distortion is not necessarily equal to the apparent or required static distortion). Although, we note that this may no longer be true if static and dynamic distortions are orthogonal; for example in the SLO desinusoiding error is

produced in the horizontal direction, whereas eye movements result in distorting functions that change only in the vertical direction.

Some of the limitations discussed above may be overly pessimistic and not detrimental in practice. In the field of AO-SLO imaging, any static distortions are expected to be less obvious than modelled here (e.g. worst case in Fig 6A). In addition, most groups employ various strategies to identify frames which appear minimally distorted for the purpose of providing a reference frame for image registration; if similar heuristics are applied to discard overly distorted frames before attempting the dynamic recovery procedure, residual errors are likely to be relatively low compared to the modelling undertaken here where all frames were included.

For retinal imaging systems adopting a wider field of view than a typical adaptive optics design, distortion is likely to be far greater as discussed in the Introduction. As noted in that section, there is a lack of experimental evidence regarding the degree of distortion imposed by the eye in conjunction with typical imaging systems. However, our method can be used to redress this lack of evidence, being applicable to data acquired from any ophthalmic imaging device. If the degree of distortion calculated is found to be substantial, the proposed algorithm can then be implemented in the image processing software of the particular device to correct image data saved in the future, and can also be applied *post hoc* to correct previously acquired data if multiple frames were recorded.

One limitation that is likely to be encountered in practice is the processing resources required for the algorithm. As described above, we solved for a 200x200 pixel patch of the image array rather than the full array. Some clipping was warranted to avoid areas of missing information due to eye movements, however, the major reason for using the smaller region of interest was that larger array sizes were not solvable on a system running with 16 GB RAM. Processing time was also several minutes for the 200x200 patch (not including image registration). Some of these issues should be addressable through more refined implementation of the algorithm, e.g. the least squares solution should be parallelisable, which should improve processing time by use of GPU computing, and also permit the use of hard disk storage to supplement RAM requirements. Where there are no rotations present, x and y solutions can be derived independently which also reduces memory requirements. However it should be noted that, in the presence of rotation, the least squares minimisation function typically converged more quickly on its solution.

In the meantime, a more pragmatic solution could be to divide the detector up into "tiles", which would be made to partially overlap to appropriately anchor the absolute offset applied. Another option to reduce computing requirements is to simply bin the registration data that is input to the algorithm, since typical distorting functions are unlikely to change to a significant degree between adjacent pixels. The solution obtained could then be upsampled and applied to the full-sized array. This solution is demonstrated in the example supplied in S1 File. Another option still could be a pyramid approach, where lower frequency information is solved first, from sub-sampled data, before turning to progressively higher frequency information sampled at greater resolution.

It should be noted that the general nature of our simulations suggest that the proposed method could be applied to imaging systems of arbitrary type, not just the eye. As long as it is possible to perform diffeomorphic registration between acquired images, the method should be applicable to any imaging system which suffers from static distortion and where it is possible (or, as in the eye, unavoidable) to pan the scene obliquely across the image array.

Finally, the method advanced here has application beyond simple imaging, to understanding the evolution of visual systems and machine vision. It is intriguing to wonder whether the small, incessant movements made by the fixating eye may have evolved, in part, due to their utility in calibrating the visual scene. This would be a useful ability for an eye to develop due to

its high optical power and non-regular arrangement of receptor elements. Previous work has described how comparison of the visual scene after small movements of the eye/head might be used to iteratively arrive at a satisfactory calibration for such causes of distortion [33]. The present work extends such thinking, providing an analytical solution that can be solved with only a handful of samples of the visual scene. We have also shown that the method can be augmented to handle temporal variations, which could arise due to non-synchronous firing of neurons or due to random movement within the scene, as occurs for example by shimmer arising from the ground on a hot day.

## Supporting information

**S1 Fig. Image stack showing combined static and dynamic distortion.** Example shows 100 contiguous frames from the cluster of 1000 frames corresponding to Fig 8A.
(TIF)

**S1 File. Software and example data to demonstrate use of the DIOS algorithm.** Code written in MATLAB R2017b (The Mathworks, USA) and requires the Image Processing Toolbox. Users should run the file named "dios_runMe.m".
(ZIP)

## Author Contributions

**Conceptualization:** Phillip Bedggood, Andrew Metha.

**Data curation:** Phillip Bedggood.

**Formal analysis:** Phillip Bedggood.

**Funding acquisition:** Phillip Bedggood, Andrew Metha.

**Investigation:** Phillip Bedggood.

**Methodology:** Phillip Bedggood, Andrew Metha.

**Project administration:** Andrew Metha.

**Resources:** Andrew Metha.

**Software:** Phillip Bedggood.

**Supervision:** Andrew Metha.

**Validation:** Phillip Bedggood, Andrew Metha.

**Visualization:** Phillip Bedggood.

**Writing – original draft:** Phillip Bedggood.

**Writing – review & editing:** Phillip Bedggood, Andrew Metha.

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
