## [Decision Letter · Decision Letter 0]

3 Feb 2021

PONE-D-20-39508

Towards distortion-free imaging for the moving eye

PLOS ONE

Dear Dr. Bedggood,

Thank you for submitting your manuscript to PLOS ONE. After careful consideration, we feel that it has merit but does not fully meet PLOS ONE’s publication criteria as it currently stands. Therefore, we invite you to submit a revised version of the manuscript that addresses the points raised during the review process.

We look forward to receiving your revised manuscript.

Kind regards,

Yuhua Zhang

Academic Editor

PLOS ONE

Journal Requirements:

2. If materials, methods, and protocols are well established, authors may cite articles where those protocols are described in detail (for instance here, Bedggood P, Metha A. De-warping of images and improved eye tracking for the scanning laser ophthalmoscope. PloS one. 2017;12(4):e0174617), but the submission should include sufficient information to be understood independent of these references

(https://journals.plos.org/plosone/s/submission-guidelines#loc-materials-and-methods)

Reviewers' comments:

Reviewer's Responses to Questions

**Comments to the Author**

1. Is the manuscript technically sound, and do the data support the conclusions?

Reviewer #1: Partly

Reviewer #2: Yes

2. Has the statistical analysis been performed appropriately and rigorously? 

Reviewer #1: N/A

Reviewer #2: No

3. Have the authors made all data underlying the findings in their manuscript fully available?

Reviewer #1: No

Reviewer #2: Yes

4. Is the manuscript presented in an intelligible fashion and written in standard English?

Reviewer #1: Yes

Reviewer #2: Yes

5. Review Comments to the Author

Reviewer #1: Review of PONE-D-20-39508

The authors demonstrated a concept of a method for correcting systematic distortions in scanning-based ophthalmic imaging instruments. Although only corrections of simulated distortions are presented, the concept itself might be an interest of the field of ophthalmic imaging.

However, the principle of the method is not well written, and all simulation parameters are not well organized. There are some questions in the result of correction for systematic and dynamic distortions (LINE 418-479). These points are required to be addressed before publication.

- Classification of distortions

The word "systematic distortion" sounds to me that static distortions due to an imaging instrument. In fact, the authors use "dynamic distortions" as a counterpart. But then the "pseudo-systemic distortion" (LINE 27) seems odd. Because "DIOS" and "dynamic restoration methods" are image-based heuristics, introducing another classification based on the type of image distortions, such as "static image distortions" and "dynamic image distortions," may help clarify the discussion of results.

-Methods

--Characteristics of imaging modalities

The authors did the proof of concept of their idea, DIOS. The images without distortions are used as ground truth, and scanning ophthalmoscope images are simulated by distorting them. To obtain the "original" ground truth images, the authors used a flood-illumination AO ophthalmoscope.

The main concern is whether the different imaging characteristics between flood-illumination-based and scanning-based ophthalmoscopes affect the performance of the presented method.

For example, speckle appearance might be different between them. The images of the flood illumination AO ophthalmoscope exhibit less speckle. However, an image of single frame of scanning ophthalmoscope may express more speckle.

For the readers to properly understand the possible difficulties, these points should be discussed, and the details of the system should be described, although the previous authors' paper [10] is cited.

--DIOS principle

Overall the principle of DIOS is quite difficult to understand. Equations are not well written, and there is no consistency among equations, figures, and tables.

--- Line 96

What does the symbol "T" mean? Although the description appears later (Line 136), it should be explained where the symbol first appears.

---Line 117

What does mean "the reference image (xR)" and "the moving image, xM"? According to the description around Equation 1, it might be the locations of the same object in the reference image and the moving image.

---Line 119-120

The numbers (2, 1, 0, 3, -3) suddenly appear. Please explain the details. Do the authors use "image 1" as "Reference image" and "image 2" as "Moving image"?

---Line 124

From two formulae ("Reference image" and "Moving image"), Equation 1 cannot be derived. The symbol "T" should disappear. Perhaps the authors should discriminate it for the reference image and moving image as "TR" and "TM".

---Figure 1

Figure 1 is unclear. What is the vertical axis? What do the labels "f(x1)", "f(x2)", and "f(x3)" mean? What do the circles mean?

---Line 215

The description of "anti-aliasing" is too short. Please describe more details, or please cite literature if there are.

The reason why the anti-aliasing can achieve a precision similar to upscaling is unclear.

---Equation

Where is Equation 5?

--Line 312

What are 2D sine functions? Equation 6 describes two 1D sine functions. Do the authors mean that both Dx and Dy are distributed in the xy-plane? If it is, Equation 6 should be corrected.

-Results

--Systematic distortion, dynamic distortion, and eye movements

There are many parameters in simulations of distortions and eye movements. It is difficult to follow the logic and interpret the results.

For each result, all simulation parameters and constraints must be explicitly described; otherwise, the results cannot be interpreted correctly. There are many ambiguities in the current manuscript. For example, the constraints to translation (eye movements simulation?) described from the Line 400 to Line 404 are commonly applied to all results?

In my opinion, the parameters and constraints of the simulation should be listed in the "Methods" section since they are the parameters of the "Verification Method."

--Systematic and dynamic distortion

As shown in Figs. 7 and 8, and discussed by the authors, there will be the interaction between distortion and eye movement. The authors conclude that the combination of the presented method (DIOS) and the previous method (dynamic recovery procedure) correct the distortions well.

This is a surprising result because the principle of DIOS assumes a common distortion (f) among image pairs. Did the authors apply common within-cluster eye movements for all clusters? Why is only one image (Fig. 8F) presented? What about other 5 images after the DIOS procedure?

If the authors' insight is correct, the discrepancy of the solution from the set systematic distortion is coming from the interaction between the distortion and eye movements. Then, the solution for correcting distortions depends on eye movement.

In order to support their conclusions, the authors need to simulate independent random intra-cluster eye movements and compare the similarity between the final corrected images with DIOS derived from each cluster.

Reviewer #2: This paper describes a method for correction of static distortion in an imaging system. This was a surprise to me, after having read only the title and the first sentence of the abstract, which led me to believe that the paper would treat the dynamic distortion caused by the interaction of intra-frame motion of the eye with a scanning system. In fact, this paper does not address that particular problem at all. For that reason, I would retitle the paper, and rewrite the abstract, possibly eliminating mention of dynamic distortion in the abstract, but in any case not leading with it!

It is really not clear to me that the method as presented has anything to do with the eye or ocular imaging at all. Clearly the authors hope to apply it to that domain, and have used high-magnification retinal images as their test images in their simulations. But the synthetic distortions that are corrected are arbitrary sinusoidal warps, that to my knowledge are common distortions in any sort of imaging system, ocular or otherwise.

Having reread the abstract, it is not obvious that I have fully understood the paper, as the sentence "the method successfully handles torsional movement of the eye" seems to go beyond correction of a static distortion. But I will leave that aside for the moment, and comment on the paper as I currently understand it.

In the third paragraph of the introduction, the authors mention several factors that can lead to "systematic distortion" (which I referred to above as "static distortion"). What is the magnitude of these problems in actual instruments? Were the authors led to this problem because systematic distortion is a problem in their own instrument?

The authors liken their problem to calibration of the intrinsic camera parameters in computer vision, and refer to "self calibration," with a number of relatively recent references given. I would like to draw the authors' attention to a 1989 paper by Maloney & Ahumada (Maloney, L., & Ahumada, A. J. (1989). Learning by assertion: A method for calibrating a simple visual system. Neural Computation, 1, 387-395) that addresses a related problem, namely how the brain "calibrates" itself to compensate for disarray in the positions of the photoreceptors. The represents an extreme form of systematic distortion, in which each input pixel is displaced in the imaging plane by a random amount. Their solution works by imposing a condition they call "translation invariance," which states that the image of a translated scene should be an equivalent translation of the image of the original scene, or I( T(s,delta) ) = T( I(s), delta ). If I have understood the present paper correctly, the method describe employs the same principle. I think the fact that I am not 100% sure means that the exposition could be clarified somewhat; I suggest starting out with a clear statement of the principle behind the solution (whether it be translation invariance or something else), before delving into the details. The original Maloney & Ahumada work assumed the motion generating the translations was known exactly (perhaps using an efference copy of the eye movement signals, or proprioception); later work explored the performance of the method using imperfectly known motion estimated from the raw uncorrected images themselves, which seems analgous to the problem of calibrating an ocular imaging system.

I did not find the one-dimensional example particularly helpful. Writing the equations in vector form can make them as concise as a 1-D solution, but then you would be done.

Here is what I would ideally like to see: apply the method to a more realistic set of synthetic images, generated by taking a large retinal image as ground truth, and then simulating capture through an imaging system containing realistic distortion as might be encountered in practice. First do it for a flood-illuminated system, so that dynamic distortions are not an issue; then simulate dynamic distortions in a scanned system by simulating sampling the image sequentially in time while it is moved according to a realistic eye movement signal (with and without torsion).

It is hard to know how to interpret the results shown in figure 7; can error bounds be put on the solution? Some statistics might help here.

-Jeff Mulligan

Minor points:

line 50: I would have thought that in the computer vision context, "rotation" of the camera includes pan and tilt, which for small angles translates the image, but is properly a rotation. To use this for camera calibration, it is necessary for the axis of rotation to pass through the optical nodal point, which requires some care.

line 55: "To our knowledge none of the methods advance have treated the case of dynamic and systematic distortions together..." - this certainly leads one to expect that the authors will treat them together, but that does not seem to be the case.

line 65: It seems the word "not" may be missing?

line 79: "... such any one of the ... images " is the word "that" missing after "such"?

Table 1: The final row appears to be a map from the first "image" to the second, but the legend states the opposite.

line 190: A little more explanation of the mathematics might be appropriate here. Matrix A is known and is sparse, but has dimensions N x M*N... Apparently the inverse matrix is really a pseudo-inverse? The description of lsqminnorm sounds like it is applied to under-constrained problems... But this problem is not under-constrained?

line 258: perfect registration is assumed; How robust is the method to registration errors - isn't that important?

Equation 6: Why is Dx only a function of x, and Dy only a function of y? (compression, but no shear) This appears not to be the case at line 324, where Dx(y) and Dy(y) are referenced?

Line 459: The polygons are not that easy to compare visually - is it not possible to come up with a quantitative measure?

Line 498: Why can't the magnitude of the distortion be solved? Is this just saying that there is an unrecoverable constant rigid transformation (like a prismatic shift)?

Line 543: Instead of referring to data in figure 7 by color, state what the different colored curves represent.

Line 546: It is assumed that eye movements are negligible compared to the speed of the fast scan in an SLO. This may not be true for saccadic velocities.

6. PLOS authors have the option to publish the peer review history of their article (what does this mean?). If published, this will include your full peer review and any attached files.

Reviewer #1: No

Reviewer #2: **Yes: **Jeffrey B. Mulligan

---

## [Author Response · Author response to Decision Letter 0]

2 Mar 2021

Please see attached document "Response to Reviewers.docx".

---

## [Decision Letter · Decision Letter 1]

20 Apr 2021

PONE-D-20-39508R1

Towards distortion-free imaging of the eye

PLOS ONE

Dear Dr. Bedggood,

Thank you for submitting your manuscript to PLOS ONE. After careful consideration, we feel that it has merit but does not fully meet PLOS ONE’s publication criteria as it currently stands. Therefore, we invite you to submit a revised version of the manuscript that addresses the points raised during the review process.

While new concerns were raised in this round of review, particularly by reviewer 2, i.e. eye position related distortion, that may help further improve the simulation, you do not have to agree to all the points. A balanced discussion of the potential limitation of the simulation work may help address the review concern.     

We look forward to receiving your revised manuscript.

Kind regards,

Yuhua Zhang

Academic Editor

PLOS ONE

Reviewers' comments:

Reviewer's Responses to Questions

**Comments to the Author**

1. If the authors have adequately addressed your comments raised in a previous round of review and you feel that this manuscript is now acceptable for publication, you may indicate that here to bypass the “Comments to the Author” section, enter your conflict of interest statement in the “Confidential to Editor” section, and submit your "Accept" recommendation.

Reviewer #1: (No Response)

Reviewer #2: (No Response)

2. Is the manuscript technically sound, and do the data support the conclusions?

Reviewer #1: Yes

Reviewer #2: No

3. Has the statistical analysis been performed appropriately and rigorously? 

Reviewer #1: N/A

Reviewer #2: No

4. Have the authors made all data underlying the findings in their manuscript fully available?

Reviewer #1: Yes

Reviewer #2: Yes

5. Is the manuscript presented in an intelligible fashion and written in standard English?

Reviewer #1: Yes

Reviewer #2: Yes

6. Review Comments to the Author

Reviewer #1: The authors extensively addressed the reviewers' concerns and the manuscript is well revised.

I only have a few concerns.

- Eye motion simulation

According to the details of the description of the presented simulation, the authors assume that the eye motion (full-field movements + dynamic distortion) has no variation in the horizontal component during single image acquisition (Dx(y) is a constant).

Even if there is a sinusoidal horizontal dynamic distortion, it is not a realistic simulation, but I think it is better to remind readers of this point.

- Comparison of 6-cluster outputs with a ground truth

The authors described the similarity between the outputs and a ground truth that "This can be further verified by the similarity with the ground truth image of Fig 8C (correlation = 0.991; recovering the images drawn from other clusters gave near-identical results) and ..."

I suggest describing the range of correlation coefficient to support their claim.

- Fig 8

"... running on the six outputs of dynamic recovery (one for each cluster), examples of which were shown in B and D."

Aren't they "D and E"?

Reviewer #2: In general, the authors have made a good effort to respond to the concerns raised by the original reviews.

I have one serious concern that did not occur to me when I read the paper for the first time a few months ago: the method explicitly solves for "static" distortion that is the same on every frame. It does this by comparing the positions of features in pairs of retinal images, obtained with the eye in a different pose. The method must assume that either the stationary eye does not contribute any distortion itself, or that whatever distortion it does contribute is constant (does not depend on eye position). I am doubtful that either one of these assumptions hold; in fact, the authors themselves mention distortion contributed by the eye at line 57. This really needs to be addressed. The simulations do not consider distortions that depend on eye position.

I am trying to imagine what would happen when a distortion-free imaging system is used to acquire retinal imagery from an eye exhibiting position-dependent distortion. I haven't been able to completely wrap my head around it, but I am confident that the result will not be an undistorted image of the retina, because the assumptions of the algorithm are violated. In practice, one approach to this problem could be to try to calibrate the instrument without an eye, by placing a calibration target at the focal plane of the device and moving it. Then when trying to estimate the (possibly position-dependent) distortion added by the eye, the instrument distortion could be factored out.

Before moving to my comments on the revised manuscript, I would like to respond to a few of the responses to my original review:

"The problem cannot be vectorized per se, although the distortion maps are represented by column vectors. At any rate, Eq. 2 could not be made any more concise." I certainly agree with the second statement, Eq. 2 is great as it stands, and should be equally true for the two-dimensional problem of interest. One equation that I would like to see in vector form is Eq. 3, which is written out as two equations, one for the x coordinate and another for the y coordinate. The x and y coordinates could be subsumed into a position p, and equation 3 could then be written in a coordinate-free way as:

f(p_R) - f(p_M) = p_M - R p_M - T

where R and T are the rotational and translational components of the affine transformation. This form still requires the definition of an origin, i.e. the location of the zero-length position vector that is the center of the rotation. As far as I can tell, this equation should be the same as equation 1. I really think that it the method can be described a little more clearly then the one-dimensional example will not really add anything.

Further on, it appears that there was a typo in my original review, as evidenced by the authors' response, "This comment is ad odds with the reviewer's comment above, with which we agree, that 'the synthetic distortions that are corrected are arbitrary sinusoidal warps, that to my knowledge are common distortions in any sort of imaging system'." What I meant to say in the original review is that to the best of my knowledge these are NOT common distortions in any sort of imaging system. I apologize for my sloppy proof-reading. That being said, I grant that any arbitrary distortion function can be synthesized from sinusoidal warps via Fourier series, so this doesn't seem like a critical issue; as the sythetic distortions appear to be quite large (possibly larger than what might be encountered in practice), if the algorithm can handle them then it will probably work well in practice. Still, it might be nice to test with a circularly-symmetric distortion function, similar to what might be encountered with a lens.

In their response to another point, the authors state that, "there is a lack of experimental evidence regarding the degree of distortion expected in real eyes." I would like to note that there is a relationship between geometric distortion and optical aberrations, and there are databases of wave aberration functions measured on collections of eyes, from which it might be possible to make reasonable estimates of the geometric distortions encountered in real eyes. This illustrates a problem with the approach, however, which is perhaps unique to the problem of ocular imaging: instrument-related distortions can safely be assumed to depend only on the sensor pixel location. Eye-related distortions, on the other hand, maybe unique for each different pose of the eye. This seems to me to be a possibly-fatal flaw with the whole approach, given that the method relies upon eye movements to produce the image shifts, and assumes that the distortion function is constant and locked to the sensor. This problem must be addressed, if only by estimating the eye-related distortions from aberration data, and showing that they are insignificant. It also seems possible that they might become significant for highly-aberrated eyes, and this should be considered also.

I have a serious concern with a statement made on line 473: the authors demonstrate that when only two images are input, which are vertically offset, that the horizontal distortion is not recovered with as much accuracy. The authors conclude that this can be fixed by shifting obliquely. This does not seem intuitively correct to me; it should be possible to formulate the problem in a coordinate-free way, in which case I would expect that larger errors would be encountered in the dimension orthogonal to the shift, regardless of what direction that might happen to be. For an oblique shift, the largest errors should be found in the orthogonal oblique direction. This is just my intuition, of course, but it seems highly unlikely to me that the coordinate axes are special, and this seems even more critical when this principle has been immortalized as part of the name of their procedure.

Detailed comments:

Line 65: It is stated that "Distortion is likely to be even higher ... when constructing montages of the retina or imaging with very wide field devices." In this case, it is not clear to me that the problem is distortion, so much as the fact the the retina is not flat, so any attempt to create a montage by translating and registering flat images is doomed to have errors, even in the absence of distortion. The projection of a spherical surface onto a planar image sensor could produce the appearance of distortion when in fact it is an inherent property of the object being imaged, not the imaging system.

Line 108: "... trial-and-error may be used to modify the system's calibration function ..." - Note that in reference 33 Maloney & Ahumada do not use "trial-and-error," but rather the Widrow-Hoff learning rule, similar to back-propagation as used in neural networks.

Line 148: The text says that the letter "H" is highlighted in red, but it does not appear so in my review copy. Consider depicting this in a way which will still work when printed in black-and-white.

Line 159: Xm is referred to as the "moving image", but wouldn't it be more accurate to call it the "translated image" or "moved image"?

Line 160: Even though it is somewhat clear from context, it would be good to define the symbols used in these equations. It is kind of confusing, because it is stated (line 165) that the expressions that are equated to x_0 "refer to 'world' co-ordinates", but clearly x_R and x_M refer to sensor co-ordinates. It would be better not to use the same symbol (x) to refer to values in different domains.

Line 168: The symbol D_M,R is said to represent "the output of pixelwise registration between the two images," and is therefore a vector (with dimensionality equal to the number of pixels), while T_M,R is "the full-field translation of the moving image relative to the reference image," which in this example is a scalar or 1-D vector. In the case of 2-dimensional images, if T is allowed to be an arbitrary rigid transformation (e.g. translation + torsional rotation), the T would need to be different for every pixel and would have the same dimensionality as D. This should be clarified. The authors rejected my suggestion to present the equations in vector form, but I still encourage them to adopt type-setting conventions that help indicate the types of various entities, such as bold non-italic lower-case letters for vectors. Perhaps in this case D and T are really functions of x? This needs to be fixed up so that a reader could easily implement the algorithm themselves from the description.

Line 193: The corresponding pixels are denoted x_1 and x_2, while earlier the were x_R and x_M. These should be consistent.

Line 217: Why are these equations relegated to a figure instead of typeset in-line, as was done for the previous equations?

Figure 2: There is something that I don't understand about this: x_1, x_2, etc, which label the columns of A, refer to locations on the sensor, while the vector f is the unknown distortion map that we wish to obtain. Why should there only be rows relating successive locations, such as x_1 and x_2, x_2 and x_3, etc? It seems like x_1 could correspond to any of the other x's for an appropriate motion. This doesn't make a lot of sense to me.

Equation 3 (line 292): T represents an affine transformation matrix, but it is written as a function with two arguments. In equation 4, there appear to be 3 parameters (theta, T_x, and T_y). What are the two arguments in equation 3?

Line 330: "Ground truth" images were obtained from a flood-illumination ophthalmoscope, and subsequently distorted with known distortions. While mostly irrelevant for the work presented, I am wondering whether the authors applied their method to solve for the (hopefully small) distortions in their ophthalmoscope?

Line 439: It is not at all clear to me that the static and dynamic distortions can be treated independently; it seems likely to me that the presence of static distortions will introduce small errors in the estimation of the dynamic distortions, although these will probably be quite small for small movements, and slowly-varying static distortions. This could be explored in simulation.

Line 600: "The method is blind to the absolute value of the distorting function..." - I think that "absolute value" is not the best term to use here. A constant translational shift is given here as an example, but isn't that in fact the only component of distortion that falls into this category? I would rephrase it to say that the method is blind to prismatic shifts, or translations that are constant over the whole image. On second thought, I think that this also includes uniform magnification/minification; if this is true, then why not mention it as well?

Lines 605-611: I'm not sure what is meant by "'islands' of connected pixels". I think what might be meant here is that however many images are collected, there need to be some that place a register-able feature at each image location. Obviously if the target (retina) has featureless regions, then there will be nothing to register and therefore no data to input to the algorithm. But with normal eyes, I don't see how this would ever arise in practice?

line 618: Another factor that could produce intensity differences between frames is vignetting produced in certain poses of the eye.

lines 622-634: The paragraph discusses a two-step procedure for dealing with simultaneous static and dynamic distortions. It is not clear to me that static distortions won't introduce biases into the solution for the dynamic distortions, and that these biases won't in turn produce error in the DIOS process. The way that I conceptualize the problem of dynamic distortion is that there is a true underlying undistorted image, with N_rows x N_cols scalar parameters (larger than a single acquired image, as this represents the correctly registered mosaic of all of the images), and three pose parameters for each frame, parameterizing the eye pose (rotation). Each image contributes M_rows x M_cols data points (the pixel intensities in a single acquired frame), and 3 additional unknowns (the pose parameters). So it is clear that as we add images, the number of inputs grows much faster than the number of unknowns. But I don't know of a practical way to perform the optimization to obtain the solution (although there certainly may be one). Existing methods might provide a good initial solution that could be further refined. Incorporating static distortion into this model adds an additional 2 x M_rows x M_cols parameters (the distortion translation vector for each pixel), but as this is the same for every frame it does not increase as frames are added, it just makes the optimization a little more complicated. Even if it is too slow to be useful in practice, if such a method could be used to generate a solution, I would have more confidence in it than sequential optimization of sub-problems, and it would be.

The problem discussed in the preceding paragraph is mentioned by the authors themselves at line 643, where they say, "It is noteworthy that although [sic] the residual static correction obtained was not simply equal to the underlying static correction applied." I'm not sure what this means, is the "underlying static correction applied" the same as the synthetic distortion applied, i.e. ground truth?

Lines 688-693: The authors discuss various ways to make the computation of the solution more efficient. One approach that they may wish to consider is a progressive "coarse-to-fine" solution, where the distortion map is built up from progressively higher frequency components, kind of like solving for the Fourier components of the distortion map sequentially. The advantage is that the lower frequency components might be solved for using subsampled data (i.e., a "pyramid" approach), reducing the amount of computation required. This suggestion is close to the approach suggested at line 690, where they suggest down-sampling by "binning."

7. PLOS authors have the option to publish the peer review history of their article (what does this mean?). If published, this will include your full peer review and any attached files.

Reviewer #1: No

Reviewer #2: **Yes: **Jeffrey B. Mulligan

---

## [Author Response · Author response to Decision Letter 1]

9 May 2021

Please see attached "Response to Reviewers" document.

---

## [Editor Report · Decision Letter 2]

25 May 2021

Towards distortion-free imaging of the eye

PONE-D-20-39508R2

Dear Dr. Bedggod, 

We’re pleased to inform you that your manuscript has been judged scientifically suitable for publication and will be formally accepted for publication once it meets all outstanding technical requirements.

Kind regards,

Yuhua Zhang

Academic Editor

PLOS ONE
---

## [Editor Report · Acceptance letter]

2 Jun 2021

PONE-D-20-39508R2 

Towards distortion-free imaging of the eye 

Dear Dr. Bedggood:

I'm pleased to inform you that your manuscript has been deemed suitable for publication in PLOS ONE. Congratulations! Your manuscript is now with our production department. 

Kind regards, 

on behalf of

Dr. Yuhua Zhang 

Academic Editor

PLOS ONE